# Preventable Adverse Events in Obstetrics—Systemic Assessment of Their Incidence and Linked Risk Factors

**DOI:** 10.3390/healthcare10010097

**Published:** 2022-01-04

**Authors:** Beate Hüner, Christina Derksen, Martina Schmiedhofer, Sonia Lippke, Wolfgang Janni, Christoph Scholz

**Affiliations:** 1Department of Gynecology and Obstetrics, University Hospital Ulm, 89075 Ulm, Germany; beate.huener@uniklinik-ulm.de (B.H.); wolfgang.janni@uniklinik-ulm.de (W.J.); 2Department of Psychology & Methods, Jacobs University Bremen, 28759 Bremen, Germany; c.derksen@jacobs-university.de (C.D.); schmiedhofer@aps-ev.de (M.S.); 3Department of Gynecology and Obstetrics, Munich Municipal Hospital Group, 81545 Munich, Germany; christoph.scholz@muenchen-klinik.de

**Keywords:** patient safety, preventable adverse events (pAEs), adverse events (AEs), obstetrics, risk management, communication

## Abstract

(1) Background: Adverse events (AEs) are an inherent part of all medical care. Obstetrics is special: it is characterized by a very high expectation regarding safety and has rare cases of harm, but extremely high individual consequences of harm. However, there is no standardized identification, documentation, or uniform terminology for the preventability of AEs in obstetrics. In this study, therefore, an obstetrics-specific matrix on the preventable factors of AEs is established based on existing literature to enable standardized reactive risk management in obstetrics. (2) Methods: AEs in obstetrics from one hospital from the year 2018 were retrospectively evaluated according to a criteria matrix regarding preventability. Risk factors for preventable AEs (pAEs) were identified. (3) Results: Out of 2865 births, adverse events were identified in 659 cases (23%). After detailed case analysis, 88 cases (13%) showed at least 1 pAE. A total of 19 risk factors could be identified in 6 categories of pAEs. (4) Conclusion: Preventable categories of error could be identified. Relevant obstetric risk factors related to the error categories were identified and categorized. If these can be modified in the future with targeted measures of proactive risk management, pAEs in obstetrics could also be reduced.

## 1. Introduction

The report “To Err is Human” published in the year 2000 led to a patient safety agenda with a focus on reducing medical errors and promoting safe health systems [1]. Patient safety is defined as “the absence of preventable harm during health care and the reduction of the risk of unnecessary harm” [2]. Adverse events (AEs) are outcomes of treatments below the current expected medical standard that result in temporary or permanent harm to patients [3]. The burden of harm in obstetrics affects not only the expectant mother, but also the newborn and the future family. In addition, the obstetric team is affected and must also deal with possible medicolegal consequences of an adverse event in the future. Reducing preventable adverse events (pAEs) is the responsibility of all levels of the healthcare system, with effective cooperation and safe communication within the team and with patients being of paramount importance [4]. The reliability of healthcare also lies in the ability of staff to adapt flexibly to the changing challenges of working conditions and patients’ needs [5]. This is particularly important for one area of healthcare, namely obstetrics, because of the complex cooperation between different professional groups, the necessarily uncontrollable aspects of every birth and the active involvement of the woman giving birth during treatment.

Different health conditions of patients complicate the systematic recording and assessment of AEs [6]. The information documented in the hospital information systems (HIS), which is not primarily used to record treatment processes, further complicates the reconstruction and assessment [7]. Internationally, different approaches to capture AEs have been reported. Most studies have conducted retrospective evaluations, including the analysis of protocol and documentation systems [8,9], by considering institutionally developed quality indicators [10,11,12]. Research teams also prospectively observed the occurrence of previously defined risk factors for AEs in birth outcomes with the conclusion that more systematic and standardized approaches are needed [13].

The medical risk to the patient must be considered when assessing harm. Previous research has distinguished between person- and system-related treatment errors, and between patient- and care-related triggers (risk factors for AEs) [6,14]. Systemic triggers are attributed to organizational problems, lack of communication, differing levels of training and lack of resources, which can only be partially controlled by the hospitals [15]. The research literature does not yet provide an agreed-upon standard for defining and recording preventability. However, action is needed, as the data on the prevalence of AEs in obstetrics is still unclear and roughly estimated at 1 to 4%, about half of which are considered preventable according to studies [6,7,8,16,17]. In Germany, there is no consensus on the systematic recording of preventable adverse events in obstetrics.

This study therefore aims to provide an overview of the annual incidence of AEs and the question of their respective preventability based on case documentation in a maximum care hospital serving as an exemplary case. The following research questions are to be answered: (a) What is the annual incidence of AEs and pAEs in obstetrics in a selected hospital? (b) How can preventability be operationalized? (c) Which measures could contribute to the reduction of pAEs?

## 2. Materials and Methods

The data were collected as part of the TeamBaby research project funded by the Innovation Fund (grant number 01VSF18023), with the aim to contribute to the reduction of pAEs [18]. All case documentation of the 3351 births at the University Hospital Ulm, Department of Gynecology and Obstetrics in the calendar year 2018 were retrospectively evaluated. The hospital in which the data were collected is a Perinatal Center Level 1, which implies a hospital with the highest level of perinatal care. This requires a 24-h in-house presence of an obstetrician and a neonatologist. High-risk pregnancies and extremely premature births can be cared for in a level 1 hospital. In this study, births before a pregnancy of <36 weeks were excluded due to the increased risks from preterm birth to avoid outcome bias. Multiple births were counted as one case. This left 2865 case records to be reviewed. There were no additional inclusion or exclusion criteria.

To identify AEs, a catalogue of criteria or events was developed based on international research findings from scientific studies [7,16,19] in two project meetings and interprofessional focus groups consisting of medical physicians, midwives, and nurses. The result was a list of 56 criteria (Table 1) which are rated as undesirable. Analogous to the classification of Forster et al. [7], 30 events refer to the physical condition of the mothers, 11 to the condition of the newborns, 12 events were assigned to interventional care and 3 to the organizational area. 

Based on the defined criteria, data from the obstetric documentation system of the University Hospital Ulm, Department of Gynecology and Obstetrics were extracted by medical documentarists into a list of AEs. All available data sources were used, including the birth documentation of the hospital, i.s.h.med system; the obstetric Viewpoint system; and the handwritten birth documentation. The cases were evaluated independently by 3 physicians and midwives and classified as preventable (i.e., as pAEs) based on 6 categories for pAEs relating to potential causative patterns (organizational error, diagnostic error, medication error, peripartum therapy delay, inadequate fetal monitoring, and inadequate maternal birth position). This procedure was performed based on the case documentation (Table 2). In addition, obstetric-relevant risk factors were identified within the pAEs (Table 3).

## 3. Results

After applying the defined criteria, obstetric AEs were identified in 659 cases (23.00% of all reviewed records) using the list of criteria. A total of 88 cases (13.35% of all cases with AEs and 3.07% of all reviewed records) had pAEs. These pAEs were categorized as peripartum therapeutic delay (e.g., delayed intervention at birth, delayed intervention for postpartum hemorrhage), diagnostic error (e.g., misdiagnosis of fetal birth position), inadequate birth position, organizational error (e.g., lack of education, lack of documentation of birth process), inadequate fetal monitoring (e.g., fetal heart rate/maternal heart rate-confusion in cardiotocography (CTG), near SIDS), and medication error. Multiple pAE categories could be assigned in each of the 88 cases. Potential obstetric risk factors were then identified in each of the cases.

### 3.1. Evaluation of the pAE Categories

A peripartum delay in therapy could be detected in 39 cases (44.32%), making it the most common cause of pAEs in this evaluation. A diagnostic error could be identified in 32 cases (36.36%). PAEs due to inadequate birth position, especially back position, occurred in 30 cases (34.09%). Due to organizational errors, pAEs occurred in 29 cases (44.32%). PAEs occurred due to inadequate fetal monitoring in 16 cases (18.18%). Medication errors occurred in 2 cases (2.27%).

### 3.2. Risk Factors as Triggers of pAE

Cases with a pAE were additionally screened for common risk factors relevant to obstetrics. A case could contain multiple risk factors. Nineteen risk factors were identified (Table 3). The risk factors could be grouped into maternal, fetal, peripartum, and organizational factors. Overall, 12 maternal and 4 fetal risk factors were identified. There were 2 peripartum risk factors and 1 organizational risk factor. In cases with a pAE, in 55.68% of cases, the woman was primiparous, and in 44.32%, she was multiparous (defined as two births or more). A common risk factor was on-call duty for doctors (work shifts between 6 p.m. and 6 a.m.; 44.32%). Induction of labor was present in 43.18%, and missed due date was present in 35.23% of cases. Obesity (BMI > 30) was identified in 23.86% of cases. Premature rupture of membranes (PROM), and language barrier were detectable risk factors in 21.59% and 20.45% of cases, respectively. Maternal age above 35 years was a risk factor in 17.05% of cases, and condition after Cesarean section in 13.64% of cases. 

### 3.3. Cross-Tabulation of Risk Factors with pAE

In Figure 1, the frequencies of risk factors of the categorized pAEs are shown and highlighted in color. Categories are sorted in descending order by number of cases (see Table 2). Risk factors are also sorted in descending order by number of cases within the grouping described above (maternal, fetal, peripartum, organizational). For each field, the number of cases in the respective pAE category (row) in which the respective risk factor (column) was present is entered. Larger numbers, i.e., a larger intersection of pAE category and risk factor, are colored darker.

Thus, the risk factor of primiparous women is frequently associated with peripartum therapy delay, diagnostic error, and missed correct diagnosis of varieties in fetal positions. This is also true in slightly lower intersection numbers for multiparous women. A missed due date is also associated with the above pAE categories. Induction of labor and on-call duty can be identified as risk factors most frequently in peripartum therapy delay, diagnostic errors, and organizational errors. Inadequate fetal monitoring is most clustered with the risk factor of on-call duty. In the case of inadequate birth position, a back position was most commonly traceable.

## 4. Discussion

This was one of the first studies which evaluated birth outcomes for AEs and pAEs in Germany systematically and openly communicated its findings. Investigation of case records of birth outcomes for the occurrence of AEs showed an incidence of 23% in births after a pregnancy of > 36 weeks. Overall, the pAE incidence appeared in approximately 3% of all births in the year 2018. This is within the range of results from other studies reporting a prevalence between 1% and 4% [7,8,16]. However, there are quite a few limitations in recording pAEs: in Germany, the data deposited in HIS primarily serve billing requirements, and the actual birth history documentation follows different systematics and is sometimes only handwritten. 

In addition, hospitals use different documentation systems, which can lead to distortions in comparative evaluations. Many often-complex circumstances that may contribute to pAEs cannot be retrospectively identified or attributed in the documentation systems, e.g., communication problems within the interdisciplinary team or budget-related staffing problems. In the following, different aspects are accordingly discussed further relating to previous evidence.

### 4.1. Operationalization of Preventability

The valid recording of the preventability of AEs requires a precise definition. The selection of the present collective of expectant mothers may have a larger proportion of patients with maternal or infant risk factors who deliver disproportionately to a perinatal center of the highest level of care in a university hospital [20]. Partly, multiple risk factors (such as obesity, diabetes, maternal age) that many women bring with them make a binary classification of the preventability of AEs difficult. At the same time, this fact emphasizes the need for a standardized, transparent assessment of preventability [21], the analysis of which can contribute significantly to prevention, i.e., more adequate management of risk factors [22]. Objectives, the selection of cases studied, methods used, and measures derived vary considerably in the literature on AEs. Some studies examine only the most serious events for their causes in the sense of a root-cause analysis to develop interventions for future prevention, while others collect the incidence from all births. 

To operationalize the aspect of preventability, six patterns of errors were identified in the present work by an interprofessional team of experts: organizational errors, diagnostic errors, medication errors, peripartum therapy delay, inadequate fetal monitoring, and inadequate maternal birth position. This categorization builds on the existing literature on the systematic recording of adverse events [2,3,4,7] since an established operationalization of the “preventability” factor of an adverse event has not yet been established for obstetrics. The published literature on this topic discussed below formed the basis for the expert team. However, it also illustrates the inconsistent approach and definitions of risk management in obstetrics.

### 4.2. Systematic Recording of Serious Incidents

To measure the success of a training program to reduce AEs in the United States, Pettker created an Adverse Outcome Index (AOI) in 2009 based on an external expert view. It includes 10 serious events ranging from maternal or neonatal death to severe birth injury. In addition, an anonymous reporting system for error documentation was implemented [8]. In the year 2017, the same author proposed a systematic root cause analysis following most serious events (sentinel events), such as intrapartum maternal or infant death, severe maternal morbidity, or transfusion reactions [16,19]. Retrospective analysis systematically examines factors that contributed to respective outcome. Sentinel events are documented in a low-threshold accessible recording system to make error transparency visible. 

Preventability is assumed for impairments due to deviation from defined standards of care. This model is used for the retrospective systematic recording of serious events, in which risk factors influencing the patient are not considered [16,19]. In Germany, corresponding documentation for quality assurance in obstetrics is performed as a standard part of the perinatal surveys of the Institute for Quality Assurance and Transparency in Health Care (IQTIG) [23]. Only serious cases are surveyed. However, improving the quality of care requires the evaluation of less severe events to identify risk factors in the field and to develop interventions. Lack of communication, which the TeamBaby research project focuses on as a potential risk factor, is most frequently associated with moderate to severe cases in the present study.

### 4.3. Evaluation of Adverse Events from Clinical Risk Management

A classical British study examined the approximately 9% of reported incidents at a maternity hospital for system- or person-based triggers and then performed root cause analysis [14]. For example, person-based problems identified misinterpretation of the CTG, medication errors or delayed diagnoses, and deviations from standard operating procedures. Systemic problems included lack of equipment and staff shortages. Inefficient communication within the team was also assessed as systemic. The goal of the analysis was to develop a catalog of interventions. As in other studies, the need for a structured “reporting system” to identify management problems was emphasized [14].

A further study [9] also examined AEs reported to risk management for influencing factors, evaluating 90 obstetric cases according to a so-called “fishbone diagram” to illustrate the relationship between the triggers, occurrence, and preventability of AEs [9]. One or more possible triggers were found in 78% of cases; of these, 31% were attributed to communication problems, another 31% to clinical problems, and 18% to diagnostic problems. In 14%, patient behavior contributed to the adverse event.

### 4.4. Risk Factor Analysis

A Spanish study [6] examined intrinsic and extrinsic risk factors to assess AE incidence. They were categorized as care-related errors, medication errors, infections, interventions, or errors related to diagnosis. Preventability was classified according to a score of 1–6 (absent/complete evidence). A score > 4 constituted a pAE according to this definition. Preventability was most frequently found in the context of interventions [6]. Such an approach was not feasible with the current study but may be helpful for the future.

### 4.5. Care Management Problems

A British study from the year 2000 identified risk factors in the team, work environment, or training to prevent future errors [13]. Factors that led to errors in care management were analyzed. Because the focus was on internal risk factors, patient-dependent risks were only included under the umbrella term “condition” and were not examined. The model is mainly suitable for structured retrospective management of AEs without consideration of patient-related risk factors. However, when aiming to prevent pAEs by means of communication, such patient-dependent risks and how to manage them might be key.

### 4.6. Prospective Analysis

The most elaborate study, with 425 patients in the USA, prospectively recorded the causes of triggers that can lead to AEs. The prospective approach also identifies factors that did not trigger a documented incident. Forster et al. compiled a list of 72 so-called trigger factors before the start of the study [7]. An obstetric professional observer team documented labor processes and trigger factors in delivery rooms, differentiated by system problems and maternal, fetal, and interventional causes.

The most common trigger was system problems at 37%, followed by maternal events at 33% and fetal events/interventions at 15% each. The decision of preventability was made consensually by an obstetric expert team after detailed case analysis when the AE was predominantly due to treatment errors [8]. From the analysis, it can be inferred that communication training and/or training to optimize documentation can reduce AEs.

### 4.7. Resilient Health Care

The studies mentioned above focus on the identification and analysis of errors that have occurred to prevent their recurrence through interventions. A complementary perspective is provided by the “Resilient Health Care” (RHC) approach, which focuses on successful care under less-than-optimal working conditions, as distinct from “error counting”. RHC proposes to complement error identification (Safety I) by establishing a safety culture that is oriented toward and reinforces successful team performance (Safety II) rather than focusing on misconduct [5]. 

A recent systematic review on RHC concepts captured performance variability, resilience, effective team relationships, and well-trained professionals as predictive factors, although authoritative study tools are still being developed [24].

### 4.8. Recommendations for Error Prevention

The approaches presented distinguish between risk factors on the part of the mother or child and system-related triggers that can be controlled to some degree. Systemic constraints which may be related to all categorized preventable adverse events in this study, such as organizational error, can be caused by inadequate staffing, frequent staff turnover, or equipment that can be optimized, are not the responsibility of the task teams. The approach of resilience orientation, which integrates the everyday strengths of employees in dealing with inadequate working conditions into a constructive safety culture, is promising due to its realistic perspective. At the same time, it is appreciative and reduces the risks of “second victims” (practitioners who suffer from the effects of treatment errors attributed to them) because it relies on respectful and constructive interaction among all professional and experience groups [25]. This is especially important if staff was involved in the occurrence of pAEs, felt responsible for prevention, and due to different reasons were not able to speak up. 

In our study, we identified categories of risk factors that were associated with pAEs. The risk factors were not weighted differently, but only evaluated numerically. In future work, one approach may be to weight the risk factors differently. For certain risk factors, greater medical attention may be given to women in the presence of these already severe risk factors. It is also possible that individual risk factors are surrogate parameters for particular risk conditions. For example, the induction of labor often involves an obstetric risk.

Dealing with pAEs is an essential part of establishing risk management. The trigger list we used provides a very differentiated analysis of obstetric cases. Not only the most serious events, such as death of the mother or child, but also minor morbid events are recorded. This leads to a higher number of cases and a differentiated evaluation of risk factors. 

A first step for error prevention could be the retrospective evaluation of data with a focus on pAEs based on a defined list of criteria as well as the analysis of risk factors. Awareness of the presence of obstetric risk factors is a key approach to reducing pAEs. A structured documentation, for example, implemented by simple checklists, helps the obstetric team to identify women with risk factors and therefore at risk for pAEs. This can improve individualized risk-adapted obstetric care. 

A further approach could be a continuous training that is routinely implemented in the obstetric setting. In particular, inadequate monitoring, diagnostic and medication errors can be partly prevented with team trainings. Additionally, simulation is key to reduce pAEs, such as peripartum therapy delay, inadequate maternal birth position and other errors linked to suboptimal hand-offs, diagnostic error, inadequate fetal monitoring and medical error. All different levels and disciplines of employees need to be involved to improve interdisciplinary understanding and collaboration. 

Among other things, the studies presented refer to inadequate communication as a cause of pAEs [26]. In addition, negative communication can increase the negative effects of inadequate staffing, thus leading to more risk factors such as diagnostic or medication errors due to stress levels, overwhelmed staff, inefficiency due to no or poor hand-offs or staff who is entirely unavailable. On the other hand, communication can be a crucial resource, e.g., when promoting optimal birth positions to the pregnant women or addressing and dealing with concerns. Thus, team training with the goal of safe communication in combination with a resilience-based approach is well suited for implementation in clinical obstetrics training to reduce the number of pAEs in the future, help staff work more efficiently, ensure patient safety, and reduce the risk of patients suffering from errors or the consequences of such errors [1,2,3]. 

## 5. Conclusions

AEs are not completely controllable in obstetrics and may only be reduced to a minimum level. Systematic analysis by means of definition and empirical categorization of AEs is also a challenge. A uniform approach is not yet available, but this research aimed to take a step in this direction. In addition, standard operating procedures (SOP) are not always exclusively applied in obstetrics and can be evaluated in case of deviation. There is an immense complexity regarding the collaboration of different professional groups, the individual care of each expectant mother with possible individual risk factors and structural as well as organizational requirements. 

This study investigated the incidence and the relationship between risk factors and the preventability of AEs with a standardized detailed criteria list adapted to German obstetrics. To operationalize preventability, an approach was developed that can be used at hospitals for the retrospective survey of AEs and subsequent development of measures to reduce AEs. In the future, one of these measures can be a systematic evaluation of the respective risk factors with classification of possible preventability. This can then be incorporated into individualized risk management (e.g., more comprehensive education on birth positions and controllable measures, standardized documentation of risks and AEs, and communication training). If risk factors can be identified as controllable and thus more preventable, this may also provide the chance to reduce pAEs. This approach should be validated at further hospitals and implemented in care practice.

## Figures and Tables

**Figure 1 healthcare-10-00097-f001:**
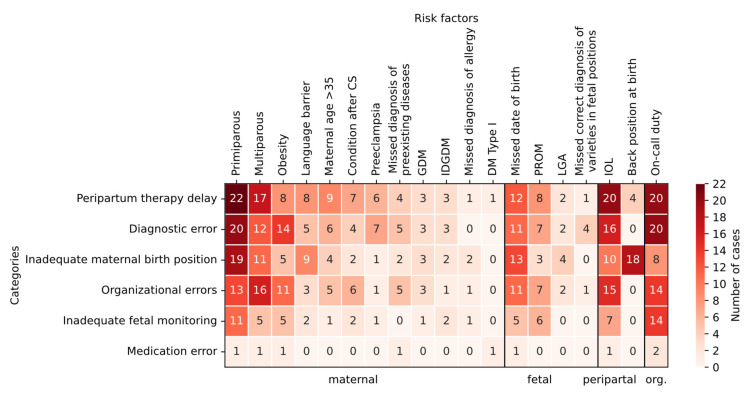
Cross-tabulation of risk factors in preventable adverse events and their overarching categories as well as their frequencies (number of cases in color). Note. CS: cesarean section, GDM: gestational diabetes, IDGDM: insulin-dependent gestational diabetes, DM: diabetes mellitus, PROM: premature rupture of membranes, LGA: large for gestational age, IOL: induction of labor; org.: organizational.

**Table 1 healthcare-10-00097-t001:** List of extracted criteria regarding adverse events; categories, thresholds, and filters for adverse events (AE).

Category	Adverse Event (AE)	Definition/Further Operationalization
Maternal	Allergy	
	Anemia	Hb < 8 mg/dL
	Postpartum length of stay	>3 days after vaginal birth
	Blood loss	>1000 mL
	Diabetic ketoacidosis	
	Disseminated intravascular coagulation (DIC)	
	Eclampsia	
	Electrolyte derailment	
	Fever	>38.5 °C
	Labor arrest	Cesarean section necessary
	Hypertension	>180/110 mmHg
	Hypotension	<90/60 mmHg
	Infection	Treatment with antibiotics
	Intubation *	
	Seizures	
	Manual placenta detachment	Non-delivered placenta
	Placental tissue after cesarean section	Curettage necessary
	Third degree laceration	
	Fourth degree laceration	
	Other laceration	Vaginal, perineal, labia
	Thyroid crisis	
	Death	
	Precipitate delivery	
	Unrecognized maternal disease	
	Unexpected re-admission	
	Uterine rupture	
	Prolonged second stage	>120 min
	Transmission to intensive care unit*	
	Placental abruption	
	Wound healing disorder	
Fetal	Near-SIDS	Near Sudden Infant Death Syndrome
	APGAR	1 min APGAR < 7
	Acidosis	Cord pH < 7.1 or base excess < −12
	Bradycardia	FHF < 60
	Birth trauma	Fracture
	Seizures	
	Meconium aspiration	
	Umbilical cord prolapses	
	Death	
	Shoulder dystocia	
	Unplanned admission to intensive care unit *	
Interventional	Transfusion	
	Failed anesthesia	
	Failed instrumental vaginal delivery	Cesarean section necessary
	Failed induction of labor	Cesarean section necessary
	Communication problem	
	Emergency hysterectomy	
	Emergency cesarean section	
	Unplanned cesarean section	
	Use of more than 1 instrument in vaginal delivery	
	Delayed intervention in case of pathological CTG	Decision-delivery time > 30 min
	Delayed intervention in case of postpartum hemorrhage (PPH)	
	Cesarean section on request	No medical indication
Organizational	Incomplete documentation	
	Medication errors	
	Communication problems	

* In the hospital where the study was conducted, women are frequently transferred to the ICU before or after delivery without the need for intubation, for example, in the case of severe hemolysis, elevated liver enzyme levels, low platelet count (HELLP syndrome).

**Table 2 healthcare-10-00097-t002:** Categories of the preventable adverse events (pAE) in the identified cases.

Category pAE	Cases	Proportion from *n* = 88 Cases
Peripartum therapy delay	39	44.32%
Diagnostic error	32	36.36%
Inadequate maternal birth position	30	34.09%
Organizational errors	29	32.95%
Inadequate fetal monitoring	16	18.18%
Medication error	2	2.27%

**Table 3 healthcare-10-00097-t003:** Risk factors for preventable adverse events (pAEs).

Risk Factors	Cases	Proportion	Category
Primiparous	49	55.68%	Maternal
Multiparous (defined as two births or more)	39	44.32%	Maternal
On-call duty	39	44.32%	Organizational
Induction of labor (IOL)	38	43.18%	Peripartal
Missed date of birth	31	35.23%	Fetal
Obesity	21	23.86%	Maternal
Premature rupture of membranes (PROM)	19	21.59%	Fetal
Back position at birth	18	20.45%	Peripartal
Language barrier	18	20.45%	Maternal
Maternal age > 35	15	17.05%	Maternal
Condition after cesarean section (CS)	12	13.64%	Maternal
Preeclampsia	9	10.23%	Maternal
Missed diagnosis of preexisting diseases	8	9.09%	Maternal
Gestational diabetes (GDM)	7	7.95%	Maternal
Large for gestational age (LGA)	6	6.82%	Fetal
Insulin-dependent gestational diabetes (IDGDM)	5	5.68%	Maternal
Missed correct diagnosis of varieties in fetal positions	4	4.55%	Fetal
Missed diagnosis of allergy	3	3.41%	Maternal
Diabetes mellitus (DM) type I	1	1.14%	Maternal

## Data Availability

The raw data from this study are available on request from the first author.

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
