# Peer review of "Preventable Adverse Events in Obstetrics—Systemic Assessment of Their Incidence and Linked Risk Factors"

_healthcare, 2022, doi:10.3390/healthcare10010097_

Round 1
Reviewer 1 Report
As a subject matter expert, I was asked to evaluate the manuscript by Hühner et al on Preventable Adverse Events (pAEs) in Obstetrics.
The authors aimed to quantify the annual incidence of preventable AEs and to offer solutions for management and prevention of potentially preventable AEs. For this purpose, they retrospectively selected all records on 3351 births from 2018 in a maximum-care hospital and analyzed all births from 37 weeks of gestation; 56 AEs were defined according to a list of criteria in alignment with international literature. Risk factors were identified and evaluated. The authors found AEs in 23% of births and potentially preventable AEs in 3% of births. The authors categorized AEs in preparation for future risk management, training, and further study.
Overall, the paper is very well written. The study objective and methodology are clear. Presentation of results and discussion are adequate. I have few minor suggestions or concerns.
Table 3 and Fig 1 and results text:
Primiparity was mentioned as one of the proportionally most significant risk factors for pAEs and was present in 56% of women. Multiparity (44%) was also described as one of the proportionally most significant risk factors. This means that pregnancy per se is 100% a risk factor for pAEs.... The authors should reflect on this point.
Primiparity may possibly be a risk factor when compared to multiparity. For mulitparity, this may well apply to women with more than 3 births. Please consider exact definition here.
In the case of on-call duty, precise definition is needed. Who does on-call duty affect? Doctors or midwives or both? If applicable, it may be better to quantify periods within 24-hour care as risk factors. E.g., there is excellent evidence that SIDS events occur primarily between 3 and 9 am. Similarly, 6-hour periods could be defined in a shift system. Induction of Labor has been labeled as a risk factor. It should be clarified that IOL may stand as a surrogate for preexisting obstetric risk, which is why IOL is performed.
It should also be discussed, if appropriate, that risk factors should be weighted differently. For certain risk factors, the event rate is unexpectedly low, e.g., diabetes mellitus, because greater medical attention may be given to women in the presence of these already severe risk factors.
Line 161: '...after a pregnancy of <36...' change to '>36....'
Line 298: '...and should not be reduced...' change into '...may only be reduced to...'
Line 306 (e.g.): Expressions like "for the first time" should be avoided. Results of similar efforts have already been presented at congresses and those expressions are not necessary to underline the value of the study.
Author Response
Thank you very much for the detailed revision of the manuscript. Please see the attachment with a point-by-point response to the comments.

Reviewer 2 Report
Manuscript ID : healthcare-1486063
It is a very interesting manuscript about preventable adverse events in obstetrics based on a retrospective analysis of 2,865 birth records in Germany. The topic is very actual.
In order to improve the manuscript, I suggest to authors the following :
- After lines 34-35: definition of adverse events, in my opinion, it is important to differentiate the burden of harm in obstetrics.
- Material and methods : please describe your inclusion and exclusion criteria. It is important to describe the level of hospital and maternity, and the standards of care, the staff, etc
- The title of the manuscript refers to a systemic assessment of the incidence and linked factors, but in Material and Method the authors refer to a project that aims to contribute to the reduction of pAEs through communication training as well as expectant mothers and accompanying persons?
- Last decade, many mits have been destroyed due to health care quality improvement. Precipitous labor was not associated with adverse events – please see : Suzuki S. Clinical significance of precipitous labor. J Clin Med Res. 2015 Mar;7(3):150-3. doi: 10.14740/jocmr2058w. Epub 2014 Dec 29. PMID: 25584099; PMCID: PMC4285060.
- It is not very clear what are adverse events in your table 1- for eg. Manual placenta detachment? I suppose in the case of the non-delivered placenta? Why is it an adverse event?
- Table 1: Intubation and transmission to intensive care unit is not the same category?
- Table 2 : I suggest to change “incorrect fetal birth position” to missed correct diagnosis of varieties in fetal positions” “allergy “ to “missed diagnosis of allergy” ; and “missed diagnosis of preexisting diseases”
- The discussion must focus more on the results of this study. “Recommendations for error prevention “(lines 269-296) is a very important subheading, but it is too general and has no relation to your data.
Author Response
Thank you very much for the detailed revision of the manuscript. Please see the point-by-point response to the comments in the attachment.

Round 2
Reviewer 2 Report
No comments